# REDUCING CLASS-WISE CONFUSION FOR INCREMENTAL LEARNING WITH DISENTANGLED MANIFOLDS

## ABSTRACT

Class incremental learning (CIL) aims to enable models to continuously learn new classes without catastrophically forgetting old ones. A promising direction is to learn and use prototypes of classes during incremental updates. Despite simplicity and intuition, we find that such methods suffer from inadequate representation capability and unsatisfied confusion caused by distribution drift. In this paper, we develop a Confusion-REduced AuTo-Encoder classifier (CREATE) for CIL. Specifically, our method employs a lightweight auto-encoder module to learn each compact class manifold in latent subspace, constraining samples well reconstructed only on the semantically correct auto-encoder. Thus, the representation stability and capability of class distributions are enhanced, alleviating the potential class-wise confusion problem. To further distinguish the drifted features, we propose a confusion-aware latent space separation loss that ensures exemplars are closely distributed in their corresponding low-dimensional manifold while keeping away from the distributions of drifted features from other classes. Our method demonstrates stronger representational capacity by learning disentangled manifolds and reduces class confusion caused by drift. Extensive experiments on multiple datasets and settings show that CREATE outperforms other state-of-the-art methods up to $5.41\%$.

## 1 INTRODUCTION

Class Incremental Learning (CIL) aims to enable deep learning models to continuously learn new classes while maintaining old knowledge. It has crucial implications in intelligent systems that require continuous evolution. For example, in an autonomous driving scenario, the system should gradually adapt to new environments, infrastructures, and traffic patterns in different countries without forgetting previous driving capabilities. A fundamental challenge in CIL is to tackle catastrophic forgetting (French, 1999; Kirkpatrick et al., 2017; Lin et al., 2024), where the performance of previously learned knowledge significantly deteriorates when the model adapts to new class instances. Existing studies are dedicated to mitigating this problem, and they primarily address the issue from three perspectives. Knowledge retention-based methods reduce the forgetting of old knowledge by preventing changes in intrinsic knowledge. Model expansion-based methods enhance adaptation to new tasks by leveraging adjustments in model parameters. Prototype-based classification methods reduce forgetting by focusing on changes in the embedding.

Specifically, knowledge retention-based approaches (Douillard et al., 2020; Chen et al., 2022; Gao et al., 2023; Fan et al., 2024) aim to discover and maintain inherent knowledge structures through regularization, thereby reducing changes in the model's intrinsic knowledge structure and minimizing the forgetting of old knowledge. Such methods impose significant constraints on the model's old knowledge, resulting in difficulties when introducing new knowledge. Therefore, model expansion-based approaches (Yan et al., 2021; Wang et al., 2022; 2023) are designed to dynamically adjust the representational capacity of a model to fit continuously evolving data. However, this type of approach typically involves a large number of parameters, and there is redundancy between the newly expanded and old branches. Prototype-based methods (Rebuffi et al., 2017; McDonnell et al., 2024; Zhou et al., 2024) construct and update prototypes, transforming the inference into a matching between features and prototypes. Such approaches are straightforward, intuitive, and require only a small number of parameters, recently showing promising prospects. However, in a class incremental

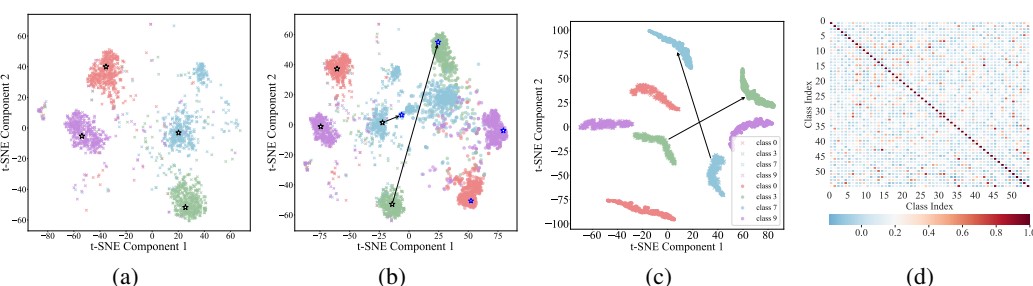

Figure 1: T-SNE visualization of feature distributions under CIFAR100 Base10 Inc10. The points marked with crosses represent features of the initial phase, while the points marked with circles indicate features of the final phases. (a) Class distributions and prototypes in the initial phase. (b) A drift in incremental learning leads to confusion in class distribution. (c) Our method exhibits reduced confusion during the drift. (d) The Pearson Correlation Coefficients (PCC) of prototypes.

learning scenario, where the distribution continuously changes, prototype-based methods are prone to occur class confusion (Yu et al., 2020; McDonnell et al., 2024).

We analyze prototype-based methods and identify two key factors that limit their performance. Firstly, real data often resides on a manifold structure in latent spaces. Observe from Fig. 1(a), Class 7 forms two distinct clusters in the latent feature space, while the prototype primarily lies in one of the clusters. This shows a single discriminative vector has limited representation capability and fails to fit the manifold distribution, leading to class-wise confusion issues. Secondly, since incremental learning cannot leverage the entire dataset, old classes often suffer from a drift after learning a new task. Fig. 1(b) shows that the positions of old classes shift severely, exhibiting significant changes in class manifolds and dispersion of features that lead to overlapping class distributions. We also calculate the Pearson correlation coefficients (PCC) (McDonnell et al. (2024)) of prototypes to verify the confusion between classes. Fig.1(d) shows that different prototypes have high linear correlations.

To address the aforementioned issues, this paper proposes a confusion-reduced auto-encoder classifier (CREATE) method as a solution. Considering auto-encoders serve as manifold learners, learning a manifold structure for each class can enhance the stability of representations while effectively capturing the essential characteristics of the categories (Bengio et al., 2013; Li et al., 2020; Zheng et al., 2022), we utilize auto-encoder reconstructions to learn class distributions. Specifically, the auto-encoder module is applied for each class to capture low-dimensional essential structures and implicitly encode the feature distribution into it, thus tackling the problem of insufficient representational capacity. Due to the overlap of the shifted representations caused by dynamically changing distributions, confusion still persists in the reconstructed representations. We further designed a confusion-aware separation loss that separates features of different classes in the class-specific latent space to mitigate the class-wise confusion.

The proposed method has the following advantages: (1) The proposed auto-encoder reconstruction modules are representation condensed and lightweight. It can effectively fit the continuously changing manifolds of data and is easily applied to existing methods. (2) It can effectively discriminate samples that suffer from distribution shifts in the feature space, thereby reducing class confusion and forgetting. Fig. 1(c) exhibits the manifold distributions in latent space, demonstrating that our method learns disentangled manifolds and reduces the class confusion caused by drift. The main contributions of this paper are summarized as follows.

• We identify the issue of class-wise confusion in incremental learning, and propose a confusion-reduced auto-encoder classifier, which uses a lightweight auto-encoder for each class to learn a compact manifold. This paradigm can exhibit a more expressive capability and effectively adapt to feature drift at the reconstruction level.

• To further reduce the confusion of drifted features, we employ a confusion-aware separation loss at the class subspace level by disentangling samples from other classes' distributions in the subspace.

- Our proposed method reduces class-wise confusion and has been validated through extensive experiments. It achieves better performance than the state-of-the-art methods up to $5.41\%$ and is easily adaptable to other methods.

## 2 RELATED WORK

### 2.1 CLASS INCREMENTAL LEARNING

Class incremental learning generally assumes that only a small number of samples can be stored for old classes, and task-id is not available in the inference phase. Existing methods can be divided into three main categories.

The knowledge retention-based methods aim to maintain the structure of old knowledge within the model and reduce knowledge variations to mitigate catastrophic forgetting. MGRB (Chen et al., 2022) constructs knowledge structure for existing classes and is utilized for regularization when learning new classes. EDG (Gao et al., 2023) maintains the global and local geometric structures of data in the mixed curvature space. DSGD (Fan et al., 2024) proposes a dynamic graph construction and preserves the invariance of the subgraph structure, which maintains instance associations during the CIL process.

The model expansion-based methods dynamically adjust model architecture to adapt new classes. For example, DER (Yan et al., 2021) expands a new backbone for each new task. The enhanced features from multiple backbone networks are concatenated for classification. FOSTER (Wang et al., 2022) adds an extra backbone for discovering complementary features and eliminates redundant parameters by distillation. Memo (Zhou et al., 2023b) expands specialized blocks for new tasks to obtain diverse feature representations.

The prototype-based methods establish a prototype representation for each task and update the prototypes in subsequent phases, classifying samples into the category of the most similar prototype in the inference phase. Some prototype-based methods use non-parametric class means as their prototypes. For example, iCaRL (Rebuffi et al., 2017) suggests the nearest class mean (NCM) classifier determines the predicted label based on the distance from the sample features to the class center. SDC (Yu et al., 2020) employs a metric loss-based embedding network and applies semantic drift compensation to adjust the prototypes closer to their correct positions. In recent years, parametric class prototypes have gained widespread use and achieved impressive performance. PODNet (Douillard et al., 2020) learns multiple proxy vectors and predicts based on the local similarity classifier. RandPAC (McDonnell et al., 2024) proposes projecting features to an expanded dimensional where enhanced linear separability of prototypes. SEED (Rypeść et al., 2024) employs one Gaussian distribution for each class and performs an ensemble of Bayes classifiers.

### 2.2 REPRESENTATION OF PROTOTYPES AND DISTRIBUTIONS

Many methods use prototypes to represent a class for classification. For example, Snell et al. (2017) formulates prototypical networks for few-shot classification. It produces a distribution for query points using a softmax over distances to the prototypes in the embedding space. Huang et al. (2022) suggests representing a class with a prototype and multiple sub-prototypes, allowing the model to better capture the diversity within the same class. Zhou & Wang (2024) proposes utilizing the centers of sub-clusters as a set of prototypes that comprehensively represent the characteristic properties. Apart from prototypes, recent methods utilize distributions for representing a category. SEED (Rypeść et al., 2024) uses multivariate Gaussian distributions to represent each class and employs Bayesian classification from all experts. This method allows for more flexible and comprehensive class representations. Considering the complex underlying structure of data distributions. Lin et al. (2024) models each class with a mixture of von Mises-Fisher distributions by multiple prototypes.

Auto-encoder structure is a type of manifold learner that can embed high-dimensional data into a low-dimensional manifold through nonlinear mapping. It is widely used for various tasks, such as anomaly detection, novel class detection, and few-shot learning. Kodirov et al. (2017) introduce a semantic auto-encoder that maps visual features to a low-dimensional semantic space, where incorporating class distribution information. Kim et al. (2019) utilizes a variational auto-encoder to learn a latent space with strong generalization capabilities for unseen classes through data-prototype im-

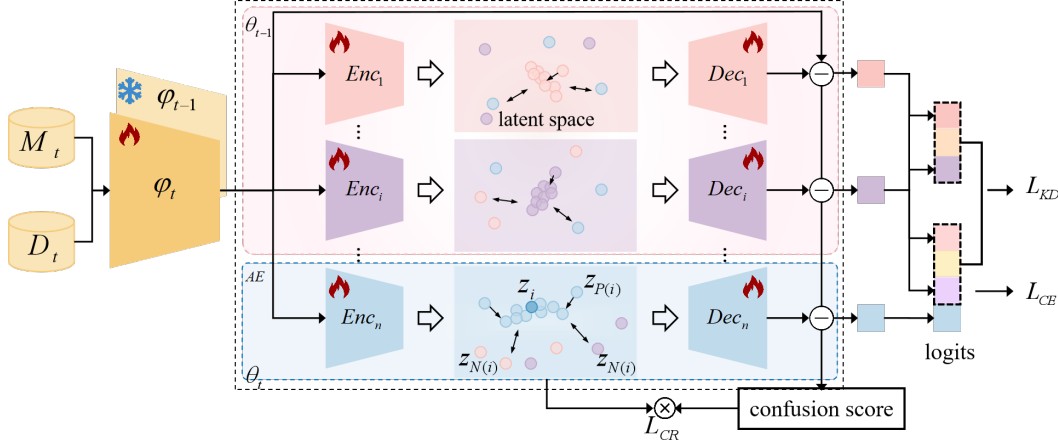

Figure 2: The overview of the proposed method confusion-reduced auto-encoder classifier (CRE-ATE). The auto-encoder (AE) learns a subspace for each class and generates a latent class distribution. Preserving the trained old AEs facilitates memory retention for old classes, while making these AEs trainable also ensures adaptability to updates in the feature extractor $\phi_t$. To further alleviate class confusion resulting from drifted distributions in class incremental learning, we employ a confusion-aware separation loss $L_{CR}$ to separate samples from other classes within each subspace.

age pairs. After training, data features are closely distributed around their corresponding prototype feature points in the latent space. Our method considers utilizing auto-encoders to enhance representation capability, as data concentrates around a low-dimensional manifold in the latent space, which is a superior characteristic for CIL to learn efficient representations of classes.

## 3 METHODS

In this section, we give a description of our confusion-reduced auto-encoder classifier in CIL 3.1. The core idea of our method is to construct a lightweight learnable auto-encoder (AE) module for each class. Preserving these trained class-wise AEs can alleviate catastrophic forgetting since they represent accurate and complete class distributions, as detailed in Section 3.2. To further mitigate the accumulated class confusion of AEs arising from distribution drift in CIL, we also propose a confusion-aware latent space separation loss in Section 3.3. Our framework is shown in Fig. 2.

### 3.1 PROBLEM SETUP

In CIL, we usually assume knowledge is not learned at once but from a sequence of $T$ tasks (phases). $\mathcal{D}_t = \{(x_i^t, y_i^t)\}_{j=1}^n$ represents $n$ samples from the task $t$. $C_t$ is the number of classes seen by phase $t$, and $n_i$ represents the number of samples in class $i$. In rehearsal-based methods, $\mathcal{M}_t$ represents the memory buffer in the $t$-th task. Therefore the training dataset in task $t$ is $\mathcal{D}_t \bigcup \mathcal{M}_t$. Note that the sets of new classes learned in different incremental tasks are mutually exclusive. The model in phase $t$ can be decomposed into feature extractor $\phi_t$ and classification module $\theta_t$.

### 3.2 LEARNING DISENTANGLED MANIFOLDS BY AUTO-ENCODER CLASSIFIER

To effectively depict class distributions and also avoid excessive computation, we consider constructing an auto-encoder module for each category so that the original features can be mapped into the corresponding subspace. The learned subspaces compress the patterns of samples into a compact and continuous low-dimensional manifold, allowing the auto-encoder to better model the distribution of the class. To adapt to new tasks and remember old classes, the previous auto-encoders are retained and kept trainable, and class-specific auto-encoders for new classes are appended. We hope the class-specific auto-encoders can identify samples of their own categories within the dataset. Therefore, we consider using reconstruction error to measure the degree of consistency between the samples and the auto-encoder subspace.

The overall framework is shown in Fig. 2. The feature extractor is utilized to obtain features, followed by a group of class-specific auto-encoders that compress and reconstruct these features for classification. Specifically, for class $i$, we construct an auto-encoder $AE_i$, consisting of an encoder $f_i : \mathbb{R}^d \to \mathbb{R}^l (d > l)$ learns a mapping that projects the original features into latent subspace, and a decoder $g_i : \mathbb{R}^l \to \mathbb{R}^d$ that reconstructs features based on the latent representations. Both encoders and decoders use a $1 \times 1$ convolutional layer with a tanh activation function. It takes features $h = \phi(x)$ as input (here, the task index $t$ is ignored for clarity), and outputs the reconstructed embeddings on each module: $\widetilde{h}_i = g_i(f_i(h))$, where $i = 1, ..., C_t$. The reconstruction error of representation $h$ on the $i$-th auto-encoder is noted as $e_i = \| \widetilde{h}_i - h \|$. When a new task arrives, auto-encoders of previous classes are reserved and keep updating to new distributions.

We use the reconstruction errors as the classification metric. On the one hand, we hope the sample on the ground truth auto-encoder has the smallest reconstruction error. This indicates that it has effectively captured the semantic knowledge and learned the distribution of specific classes. This can be achieved by minimizing the reconstruction errors to zero for samples on their ground truth auto-encoders. On the other hand, we expect samples processed by modules that do not belong to their specific classes to exhibit larger errors, indicating that mapping samples to the wrong subspaces results in significantly mismatched. Therefore, we process the reconstruction errors as Eq.1 to obtain the predicted probability of the sample $x$:

$$p_i = \frac{\exp(-\alpha e_i)}{\sum_{j=1}^{C_t} \exp(-\alpha e_j)}, \tag{1}$$

where $\alpha$ is a positive hyper-parameter used to adjust the scale of the reconstruction errors. We can see that the probability $p_i$ is negatively correlated with the distance between the reconstructed feature on $i$-th auto-encoder and the original feature. Then, we employ the cross-entropy loss function to measure the difference between the predicted probability distribution and the ground truth, which is helpful for inter-class discrimination:

$$L_{CE} = -\sum_{i=1}^{C_t} y_i \log p_i. \tag{2}$$

To mitigate forgetting of old classes, we apply the distillation loss on the logits-level, formulated as:

$$L_{KD} = -\sum_{i=1}^{C_{t-1}} \frac{\exp(-\alpha \bar{e}_i/T)}{\sum_{j=1}^{C_{t-1}} \exp(-\alpha \bar{e}_j/T)} \log \frac{\exp(-\alpha e_i/T)}{\sum_{j=1}^{C_{t-1}} \exp(-\alpha e_j/T)}, \tag{3}$$

where $\bar{e}_i$ is the logits provided by the old network. Note that the feature extractor and auto-encoders remain unfrozen during training to adapt to new tasks.

### 3.3 CONFUSION-AWARE LATENT SPACE SEPARATION

Due to the feature shift in incremental learning, confusion between classes is becoming increasingly severe, leading to catastrophic forgetting. We propose further class separation within the subspace to reduce confusion when distinguishing shifted features. This process is essential as features of different classes may distribute in similar positions after a shift, causing compressed low-dimensional features relatively similar to each other on the manifold, and so do the reconstructed features. The approximate reconstruction errors thus confuse classification.

Firstly, we measure the confusion score $s_i$ for sample $i$ expressed as follows:

$$s_i = \frac{|e_{i(2)} - e_{i(1)}|}{e_{i\max} - e_{i(1)}}, \tag{4}$$

where $e_{i(1)}$ and $e_{i(2)}$ are the smallest and the second smallest values in the error sequence $e_i$, respectively. A smaller $s$ for a sample suggests similar reconstruction errors across different auto-encoders, indicating significant confusion.

Therefore, we suggest that samples should located far from the manifold region in their non-ground truth auto-encoders' latent space. We employ a contrastive loss in the class-specific subspaces, written as follows:

$$L_{CST} = \sum_{i=1}^{C_t} -\frac{1}{|P(i)|} \sum_{p \in P(i)} \log \frac{\exp(z_i^T \cdot z_{p_i}/\tau)}{\sum_{k \in P(i) \cup N(i)} \exp(z_i^T \cdot z_{k_i}/\tau)}, \tag{5}$$

where $z_i = f_i(\phi(x))$, $P(i)$ represents the positive samples set of class $i$, and $z_{p_i}$ is the latent representation on the $i$-th class auto-encoder for samples that share the same label as $z_i$. $N(i)$ denotes the set of negative samples of class $i$.

Optimizing Eq. 5 allows us to learn more accurate manifold and reduce class-wise confusion problem. This is achieved by maximizing the mutual information between positive samples, which draws positive pairs closer in the latent space and fits a low-dimensional manifold, while simultaneously minimizing the mutual information between negative samples, thereby pushing them further away from the manifold region of the particular class.

Considering the varying degrees of sample confusion, we transpose the confusion scores into weights by Eq. 6 and get a confusion-reduce contrastive loss function formulated as Eq. 7.

$$w_i = 1 + e^{-\beta s_i}, w_i \in [1, 2], \tag{6}$$

$$L_{CR} = \sum_{i=1}^{C_t} \frac{-1}{|P(i)|} \sum_{p \in P(i)} w_i \log \frac{\exp(z_i^T \cdot z_{p_i}/\tau)}{\sum_{k \in P(i) \cup N(i)} \exp(z_i^T \cdot z_{k_i}/\tau)}. \tag{7}$$

The complete loss for this model is formulated as:

$$L = L_{CE} + L_{KD} + \lambda L_{CR}. \tag{8}$$

In summary, our proposed model addresses two issues in existing prototype-based methods: insufficient representational capacity and sensitivity to shifted features. We enhance representational capability by utilizing a group of auto-encoders to capture the unique distribution information of each class and improve class separation in the latent representation to reduce sensitivity to shifted features. Ultimately, this approach helps mitigate the confusion problem encountered in incremental learning.

## 4 EXPERIMENTS

### 4.1 EXPERIMENTAL SETUPS

**Datasets.** CIFAR100 (Krizhevsky et al., 2009) consists of 32x32 pixel images and has 100 classes. Each class contains 600 images, with 500 for training and 100 for testing. ImageNet100 (Deng et al., 2009) is selected from the ImageNet-1000 dataset, comprising 100 distinct classes. Each class contains about 1300 images for training and 500 images for testing.

**Protocols.** For CIFAR100 and ImageNet100, we evaluate the proposed method on two widely used protocols: Base0 for learning from scratch and Base50 for learning from half. In Base0, classes are evenly divided. Inc10 and Inc20 refer to tasks containing 10 and 20 classes, incrementally learning until all classes are covered. Up to 2,000 exemplars can be stored. Base50 refers to a model that learns 50 classes in the first phase, and then learns the remaining 50 classes in Inc5 mode (5 classes per task) or Inc10 mode (10 classes per task). The memory buffer is set to 20 exemplars per class. We denote the accuracy after task $t$ as $A_t$ and use the final phase accuracy $A_T$ and average incremental accuracy $\bar{A} = \frac{1}{T} \sum_{t=1}^{T} A_t$ for comparison. We use '#P' to denote the parameters count in million after the final phases.

**Implementation details.** The proposed method is implemented with PyTorch and PyCIL (Zhou et al., 2023a). Experiments are run on NVIDIA RTX3090 GPU with 24 GB. We employ ResNet18

Table 1: Last and average accuracy of different methods on CIFAR100. The best performance is highlighted in bold, while the second-best performance is indicated with underline.

| Methods | CIFAR100 B0 | | | | CIFAR100 B50 | | | |
|---|---|---|---|---|---|---|---|---|
| | Inc 10 | | Inc 20 | | Inc 5 | | Inc 10 | |
| | Last | Avg | Last | Avg | Last | Avg | Last | Avg |
| iCaRL (CVPR' 2017) | 49.52 | 64.42 | 54.23 | 67.00 | 47.27 | 53.21 | 52.04 | 61.29 |
| PODNet (ECCV' 2020) | 36.78 | 55.22 | 49.08 | 62.96 | 52.11 | 62.38 | 55.21 | 64.45 |
| WA (CVPR' 2020) | 52.30 | 67.09 | 57.97 | 68.51 | 48.01 | 55.90 | 55.85 | 64.32 |
| DER (CVPR' 2021) | 58.59 | 69.74 | 62.40 | 70.82 | 56.57 | 64.50 | 61.94 | 68.24 |
| Foster (ECCV' 2022) | 62.54 | 72.81 | 64.55 | 72.54 | 60.44 | 67.94 | 64.01 | 70.10 |
| DyTox (CVPR' 2022) | 58.72 | 71.07 | 64.22 | 73.05 | - | - | 60.35 | 69.07 |
| BEEF (ICLR' 2023) | 60.98 | 71.94 | 62.58 | 72.31 | 63.51 | 70.71 | 65.24 | 71.70 |
| DGR (CVPR' 2024) | 57.10 | 68.40 | 61.90 | 70.70 | 54.70 | 61.90 | 58.90 | 66.50 |
| DSGD (AAAI' 2024) | 63.18 | 73.01 | 67.67 | 72.91 | 63.58 | 68.14 | 65.83 | 70.02 |
| CREATE | **63.69** | **75.60** | **69.99** | **78.46** | **63.53** | **72.27** | **68.40** | **75.52** |
| Gain (Δ) | +0.51 | +2.59 | +2.32 | +5.41 | +0.05 | +1.56 | +2.57 | +3.82 |

Table 2: Last and average accuracy of different methods on ImageNet100. The best performance is highlighted in bold, while the second-best performance is indicated with underline. '#P' represents the number of parameters (million).

| Methods | ImageNet100 B0 | | | | | | ImageNet100 B50 | | | | | |
|---|---|---|---|---|---|---|---|---|---|---|---|---|
| | Inc 10 | | | Inc 20 | | | Inc 5 | | | Inc 10 | | |
| | #P | Last | Avg | #P | Last | Avg | #P | Last | Avg | #P | Last | Avg |
| DyTox | 11.00 | 61.78 | 73.40 | 11.00 | 68.78 | 76.81 | 11.00 | - | - | 11.00 | 65.76 | 74.65 |
| iCaRL | 11.17 | 50.98 | 67.11 | 11.17 | 61.50 | 73.57 | 11.17 | 50.52 | 57.92 | 11.17 | 53.68 | 62.56 |
| PODNet | 11.17 | 45.40 | 64.03 | 11.17 | 58.04 | 71.99 | 11.17 | 64.70 | 72.59 | 11.17 | 62.94 | 73.83 |
| WA | 11.17 | 55.04 | 68.60 | 11.17 | 64.84 | 74.44 | 11.17 | - | - | 11.17 | 56.64 | 65.81 |
| Foster | 11.17 | 60.58 | 69.36 | 11.17 | 68.88 | 75.27 | 11.17 | 67.78 | 76.21 | 11.17 | 63.12 | 69.85 |
| DGR | 11.17 | 64.00 | 72.80 | 11.17 | 71.10 | 77.50 | 11.17 | 62.60 | 70.50 | 11.17 | 69.30 | 74.90 |
| DER | 111.7 | 66.84 | 77.08 | 55.85 | 72.10 | 78.56 | 122.87 | 69.30 | 77.69 | 67.02 | 71.10 | 77.57 |
| DSGD | 111.7 | 68.32 | 75.68 | 55.85 | 71.76 | 77.07 | 122.87 | 69.50 | 77.20 | 67.02 | 73.01 | 80.30 |
| BEEF | 111.7 | **71.12** | **79.34** | 55.85 | - | - | 122.87 | - | - | 67.02 | 74.62 | 80.52 |
| CREATE | 14.44 | 66.58 | 77.49 | 14.44 | **74.34** | **81.37** | 14.44 | **71.42** | **79.44** | 14.44 | **77.06** | **82.43** |
| Gain (Δ) | | | | | +2.24 | +2.81 | | +1.92 | +1.75 | | +2.44 | +1.91 |

(without pre-training) as the feature extractor for both CIFAR100 and ImageNet100. We adopt an SGD optimizer with a weight decay of 2e-4 and a momentum of 0.9. We train the model for 200 epochs in the initial phase and 120 epochs in the subsequent incremental phase. The batch size is 128, and the initial learning rate is 0.1. We set the hyper-parameter $\alpha$ to 0.1 and $\beta$ to 2. The temperature $\tau$ in $L_{KD}$ is set to 2, and $\lambda$ is set to 1 for all experiments.

## 4.2 EXPERIMENTAL RESULTS

**Comparative performance.** Table 1 and Table 2 presents the comparative experiments on CIFAR100 and ImageNet100. We run four settings on the CIFAR100 and Imagenet100 dataset including both learning from scratch and learning from half, incremental learning 5 and 10 phases, and report the last phase accuracy and the average accuracy. It can be seen that our method surpasses the best results in both last accuracy and average incremental accuracy by $1.36\%$ and $3.35\%$, respectively, on average across the four settings on CIFAR100. The greater enhancements in average incremental accuracy indicate that our method ensures a steady improvement throughout the entire learning progress, rather than only in the final phase. In the learning from half setting Base50 Inc10, our method gets an average accuracy improvement of $3.82\%$ over BEEF (Wang et al., 2023) and a

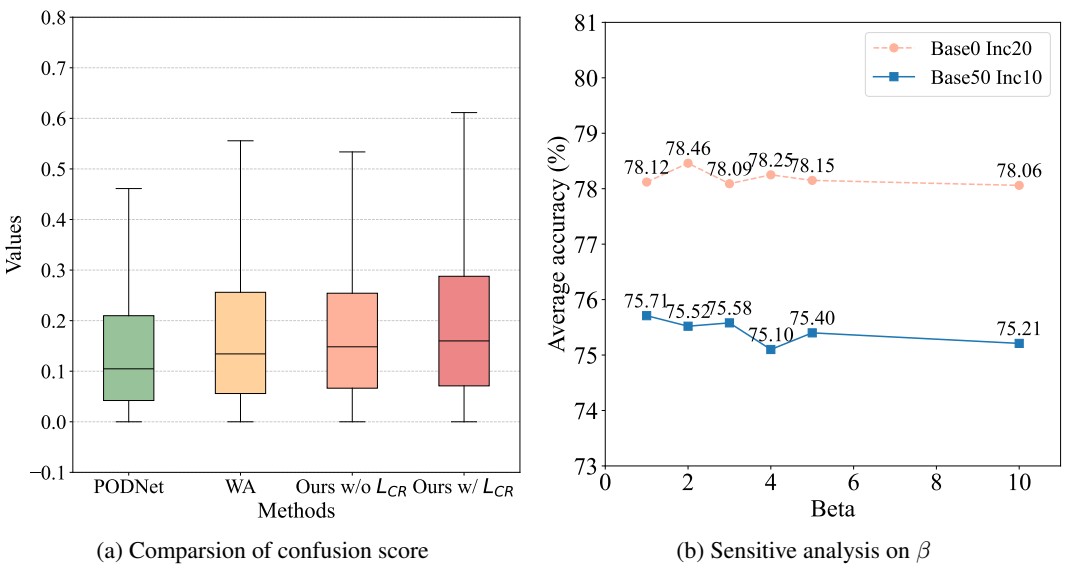

(a) Comparsion of confusion score  (b) Sensitive analysis on $\beta$

Figure 3: Analysis of confusion degree and hyper-parameter sensitivity.

final accuracy enhancement of $2.57\%$ over DSGD (Fan et al., 2024). In the learning from scratch setting Base0 Inc20, our method achieves $2.32\%$ in last accuracy and $5.41\%$ in average accuracy higher than the DSGD and DyTox, respectively.

Table 2 presents the comparative experiments on ImageNet100. Our method also surpasses existing state-of-the-art methods by a significant margin in most settings. In Base50 Inc5, our method gets an average accuracy improvement of $1.91\%$ and a final accuracy enhancement of $2.44\%$ over BEEF. Our framework can capture intrinsic manifold distribution and improve the separability of confusing features, yielding remarkable results in both Base0 and Base50 protocols.

**Parameter efficiency.** In addition to comparative accuracy, our method exhibit parameter efficiency in ImageNet100. The comparative experiments conducted on the ImageNet100 are presented in Table 2. Our proposed method consistently outperforms the state-of-the-art methods by $5.28\%$ and $1.91\%$ in the average accuracy of learning from half settings, and reduces the parameter by $88.2\%$ and $78.4\%$, respectively. This reduction highlights the superiority of our auto-encoder architecture. Our method benefits from the enhanced representational ability and reduced confusion under severe drifts. Although BEEF achieves a higher performance in the Base10 Inc10 setting, it comes at the cost of approximately ten times the parameter scale compared to our method. Our method requires significantly fewer parameters than model expansion and fusion-based approaches and achieve competitive results.

### 4.3 ABLATION STUDY

In this section, we conduct experiments to verify the effectiveness of the components. We validated the following three aspects: (1) quantitative analysis on component effectiveness, (2) class confusion reduction analysis, and (3) impact of hyper-parameters.

Table 3: Ablations study in our method. We report the accuracy of each phase and the average accuracy under CIFAR100 Base50 Inc10.

| Comp. | NCM | AEs | $L_{CR}$ | 50 | 60 | 70 | 80 | 90 | 100 | Avg |
|-------|-----|-----|----------|-----|-----|-----|-----|-----|-----|-----|
| NCM | ✓ | | | 84.80 | 75.90 | 69.94 | 65.82 | 63.03 | 62.09 | 70.26 |
| Ours-AE | | ✓ | | 84.20 | 79.75 | 76.13 | 71.26 | 68.47 | 65.31 | 74.19 |
| Ours | | ✓ | ✓ | 84.64 | 80.42 | 76.79 | 72.80 | 70.09 | 68.40 | **75.52** |

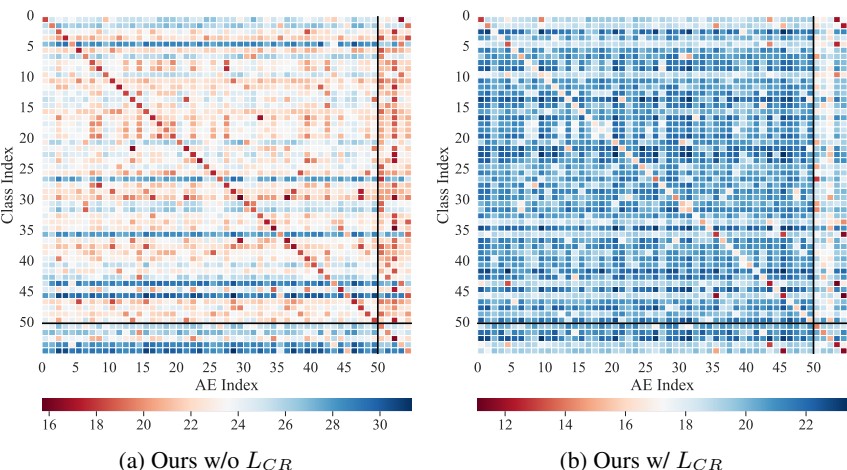

(a) Ours w/o $L_{CR}$          (b) Ours w/ $L_{CR}$

Figure 4: Reconstruction errors of misclassified data in CIFAR100 Base50 Inc5 phase2.

**Effectiveness of Components.** We conduct ablation experiments on CIFAR100 Base50 Inc10. **NCM** means predicting the labels of test samples by nearest-class-means that computing distances between their embeddings and prototypes of each class. **Ours-AE** infers labels based on class-specific auto-encoders without the $L_{CR}$. **Ours** contains both the proposed framework and $L_{CR}$. As shown in Table 3, the accuracy increases as we gradually add the proposed components. The proposed framework improves the average accuracy by 3.93% over the classical prototype-based method. The final composition of the method raises the performance to 75.52%.

**Class confusion analysis.** We draw the box plot of confusion scores for multiple methods on the test set in Fig. 3(a). The confusion score is defined by Eq. 4, where for other methods, the variable $e$ is replaced with logits. A smaller confusion score signifies a greater degree of confusion in category predictions. Our method exhibits a higher confusion scores compared to other approaches. It can be observed that both the mean value of the confusion score and the upper quartile are higher than those of the comparison methods. This indicates that our model can effectively distinguish and reduce confusion when faced with shifted features.

Fig. 4 shows the impact of our confusion-aware separation loss after learning the second phase. We find that implementing $L_{CR}$ increases the reconstruction error of the auto-encoder for shifted samples that do not belong to their respective semantic categories. Thus, $L_{CR}$ can help enhance the class separation in the latent space, thereby alleviating the degree of confusion for CIL.

**Impact of hyper-parameters.** The hyper-parameters used in the method are $\alpha$ for scaling reconstruction errors and $\beta$ for controlling samples confusion weights. We set $\alpha$ to 0.1 in all experiments to prevent overflow when taking the logarithm of the reconstruction errors. Thus, we conducted experiments on the remaining hyper-parameter $\beta$. As shown in Fig. 3(b), among $\beta$ values of 1, 2, 3, 4, 5, and 10, the average accuracy remains relatively stable in various settings.

## 5 CONCLUSION

In this paper, we propose an auto-encoder classifier to reduce class-wise confusion in incremental learning. It employs a lightweight auto-encoder module and learns disentangled manifolds for each class to represent their distribution. Moreover, it constrains latent spaces by a confusion-aware separation loss that enhances class separability. This approach addresses the problem of insufficient representational capacity and severe class confusion in the dynamic distribution-changing situation of prototype-based CIL methods. Experimental results show that our method achieves state-of-the-art performance in various scenarios.

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
