# OpenReview forum: "Reducing class-wise confusion for incremental learning with disentangled manifolds"
_ICLR.cc/2025/Conference — ICLR 2025 Conference Withdrawn Submission_

### Official Review · Reviewer_9Rkx · 2024-10-19

**Soundness:** 3
**Presentation:** 3
**Contribution:** 2
**Rating:** 5
**Confidence:** 4

**Summary:**

This paper focuses on the scenarios of class-incremental learning. To tackle the inadequate representation capability and unsatisfied confusion caused by distribution drift, this paper leverages autoencoders as the classifier in the continual learning of the model. The authors claim it captures compact class manifold in the latent space. The paper also proposes a confusion-aware latent space separation loss to deal with the feature drift.

**Strengths:**

1.	This paper is clear and easy to read.
2.	The proposed classifier based on autoencoder is novel.

**Weaknesses:**

1.	The t-SNE experiment showing the effectiveness of the proposed method (Figure 1 (a)(b)(c)) is not persuasive enough. It seems (a)(b) and (c) use different settings of t-SNE (such as different number of iterations) to get the results, because the level of the number of outliers in (a)(b) are obviously higher than that in (c), which can be reduced by increasing the number of iterations in t-SNE.
Therefore, for me, the reduced confusion cannot be inferred from this experiment.
2.	The performance of the baseline methods is questionable. For CIFAR100 B0 Inc 10, DER achieves 75.36 for the average accuracy in the original paper using ResNet18 as the backbone, this number is 69.74 in this paper.

**Questions:**

See weaknesses and below.

The cross entropy on the predicted probability produced by the reconstruction errors (eq(1)(2)) not only increases the reconstruction ability on the class-specific autoencoder for a sample, but also decreases the reconstruction ability on other class-specific autoencoders. Given that the imbalanced training set with a few exemplars and unfrozen previous autoencoders, the autoencoders for the previous classes are easily affected by the majority samples of the current task. Even though, this method achieves better results than the expansion-based methods like DER and BEEF. This leaves me a question that what superiority does the proposed method have over the expansion-based methods?

---

### Official Review · Reviewer_iW4V · 2024-10-27

**Soundness:** 3
**Presentation:** 3
**Contribution:** 2
**Rating:** 5
**Confidence:** 5

**Summary:**

This paper introduces CREATE, a confusion-reduced method for class incremental learning. A lightweight auto-encoder is designed to learn a separate manifold in latent subspace for each class. Additionally, a confusion-aware latent space separation loss is proposed to enhance the distinction of the drifted features further. Results and ablation studies also show the effectiveness of the proposed method.

**Strengths:**

1.	The paper is well-written, making it easy to follow
2.	The figures effectively illustrate the motivation and overall pipeline of the method
3.	The results from various settings, along with the ablation studies, demonstrate the effectiveness of the proposed method.

**Weaknesses:**

1.	Several recent papers[1,2] also analyze the drift of prototypes, these should be discussed in the related work section.
2.	Results on ImageNet100 B0 Inc10 are inferior to BEEF.
3.	I suggest that the authors provide a comparison of average forgetting across different methods, as this could clarify whether performance improvements stem from enhanced knowledge acquisition or reduced forgetting.
4.	In Figure 3(a), the authors should include more recent methods such as BEEF and DGR to better demonstrate the effectiveness of their proposed method compared to the latest approaches.

[1]Li Q, Peng Y, Zhou J. FCS: Feature Calibration and Separation for Non-Exemplar Class Incremental Learning[C]//Proceedings of the IEEE/CVF Conference on Computer Vision and Pattern Recognition. 2024: 28495-28504.

[2]Shi W, Ye M. Prototype reminiscence and augmented asymmetric knowledge aggregation for non-exemplar class-incremental learning[C]//Proceedings of the IEEE/CVF International Conference on Computer Vision. 2023: 1772-1781.

**Questions:**

I am curious why the encoder and decoder for previous classes are trainable during the training of later classes, as this may lead to knowledge forgetting for those earlier classes.

---

### Official Review · Reviewer_ih8W · 2024-11-03

**Soundness:** 2
**Presentation:** 2
**Contribution:** 2
**Rating:** 6
**Confidence:** 4

**Summary:**

This paper introduces an auto-encoder based method to conduct incremental learning. The proposed auto-encoder based classifier can be used to reduce class confusion in class incremental learning, which further alleviate forgetting.

**Strengths:**

(1) The proposed auto-encoder-based classifier is novel and can effectively disentangle representation of different classes, reducing the representational class confusion.
(2) The introduced auto-encoder is very lightweight, which does not introduce too much overhead during incremental learning.
(3) The proposed method is shown to have significant improvement over previous baselines.
(4) Detailed ablation studies and sensitivity analysis are provided.

**Weaknesses:**

(1) Experiments on larger-scale setting such as ImageNet-1000 is missing.
(2) The inference can be a bit cumbersome, as a representation embedding need to iterate different auto-encoders to obtain the reconstruction error.

**Questions:**

See weakness (2). Will the additional auto-encoder introduced at each learning phase hinder the scalability of the proposed method in terms of inference speed?

---

### Official Review · Reviewer_khYV · 2024-11-03

**Soundness:** 2
**Presentation:** 2
**Contribution:** 2
**Rating:** 3
**Confidence:** 4

**Summary:**

The paper highlight the issue of class-wise confusion and feature drift when using prototype-based methods in continual learning. The authors propose to use an auto-encoder classifier which learns a compact manifold for each class. The proposed method employs a confusion-aware separation loss to reduce class-wise confusion. The method improves over existing state-of-the-art methods.

**Strengths:**

1. The paper discusses the issue of feature drift in prototype-based methods.
2. The concept of using auto-encoder classifiers is nice.
3. The extensive experiments and the discussion on parameter efficiency is appreciated.

**Weaknesses:**

1. Fig 1 does not really add anything new. This is already shown in [1,3]. Fig 1 (c) shows that using the proposed method, the feature distributions are still drifting significantly. This needs to be explained in more details to highlight what exactly is the method solving.

2. While the paper analyze and introduce the problems of prototype-based methods, it does not compare with any prototype-based methods like [1,2] or with drift compensation-based methods [3,4]. The shift from prototype-based methods (which are mostly exemplar-free) in the theory to exemplar-based methods in experiments is hard to understand. The context of the motivation is completely different from the solution. This is because exemplar-based methods can also use Nearest-Mean-of-Exemplars [5] (NCM where old class prototypes can be updated using exemplars on the newly updated model) and thus the problem of drift in prototypes is resolved to a big extent. The main issue of feature drift in prototype-based methods [1,2,3,4] is primarily because of not having exemplars. Hence, the motivation of this paper and the issues does not really apply to the exemplar-based settings. I see this as a major issue of this work.  It is important to show how the method addresses prototype drift in exemplar-based settings specifically, and explain why focus on exemplar-based experiments given the initial focus on exemplar-free prototype-based methods. I would ask the authors to include comparisons with prototype-based and drift-compensation methods.

3. Claims are not validated. For instance, line 109 - “is easily adaptable to other methods.” The paper does not provide any such experiments or validations. The authors should provide specific examples or experiments demonstrating how their method can be adapted to other approaches.

4. The ablation experiments are not convincing. For instance, advanced NCM classifiers exists like FeCAM (Mahalanobis distance-based classifier which significantly outperforms NCM) [2] and Nearest-Mean-of-Exemplars [5] (specifically for exemplar-based settings) which are not compared or discussed.

[1] McDonnell, Mark D, et al. “Ranpac: Random projections and pre-trained models for continual learning.” Advances in Neural Information Processing Systems, 36, 2023.

[2] Goswami, Dipam, et al. "Fecam: Exploiting the heterogeneity of class distributions in exemplar-free continual learning." Advances in Neural Information Processing Systems 36 (2023).

[3] Yu, Lu, et al. "Semantic drift compensation for class-incremental learning." Proceedings of the IEEE/CVF conference on computer vision and pattern recognition. 2020..

[4] Goswami, Dipam, et al. "Resurrecting Old Classes with New Data for Exemplar-Free Continual Learning." Proceedings of the IEEE/CVF Conference on Computer Vision and Pattern Recognition. 2024.

[5] Rebuffi, Sylvestre-Alvise, et al. Icarl: incremental classifier and representation learning. In Proceedings of the IEEE Conference on Computer Vision and Pattern Recognition, 2017.

**Questions:**

What exactly is the "class-wise confusion" issue? This is not a commonly used term in CL. How is this different from feature/prototype drift? This definition is not clear from the paper. An illustration or formal definition of the two issues will make it more clear.

---

### Official Review · Reviewer_ADUW · 2024-11-04

**Soundness:** 2
**Presentation:** 3
**Contribution:** 3
**Rating:** 5
**Confidence:** 4

**Summary:**

To address the issues of insufficient representation capability and class confusion caused by distribution drift in existing prototype-based classification methods in class incremental learning, this paper proposes a method named CREATE. Based on a replay strategy, this method introduces an autoencoder module for each class to learn compact manifold structures. Simultaneously, to prevent class confusion, contrastive learning is employed in the latent space across autoencoders of all classes to increase inter-class distances. During classification, the reconstruction loss of sample features on each autoencoder is calculated as the basis for classification. The lightweight nature of the auto-encoder modules also makes this method feasible for use in resource-constrained environments, potentially broadening the applicability of CIL methods in the real world. Experiments demonstrate the effectiveness of the proposed method.

**Strengths:**

The paper proposes an innovative solution to the class confusion problem in class incremental learning (CIL) by introducing a CREATE classifier. This approach is original in that it leverages class-specific auto-encoders to disentangle class manifolds, which is a novel application in the context of CIL. Unlike traditional methods that primarily rely on prototypes or parameter-based strategies to combat catastrophic forgetting, CREATE uses a lightweight auto-encoder model to capture compact manifolds for each class, addressing the issue of class overlap caused by distribution drift. This paper is well-written, logically clear, and expresses the research objectives intuitively through visualization.

**Weaknesses:**

Absence of Analysis on Inter-Class Similarity:  The model uses class-specific auto-encoders and separation loss to reduce class-wise confusion, but it does not address how well it performs when classes are highly similar. In real-world applications, some classes may share overlapping features, leading to higher confusion. Providing analysis or visualizations of how the model handles such cases would strengthen the claim that CREATE effectively reduces class-wise confusion.
Class-Specific vs. Unified Reconstructor: The paper lacks a comparison between class-specific and unified feature reconstructors. Clarifying this difference, along with comparisons to [1] and [2], would better highlight CREATE’s unique advantages.

[1] Zhou, Y., Yao, J., Hong, F., Zhang, Y., & Wang, Y. (2024). Balanced destruction-reconstruction dynamics for memory-replay class incremental learning. IEEE Transactions on Image Processing.

[2] Zhai, J. T., Liu, X., Bagdanov, A. D., Li, K., & Cheng, M. M. (2023). Masked autoencoders are efficient class incremental learners. In Proceedings of the IEEE/CVF International Conference on Computer Vision (pp. 19104-19113).

**Questions:**

(1). When replay samples are involved in training, does the feature shift remain as severe? Additionally, since the feature reconstructor for previous tasks is updated while training on the current task, is there a risk of catastrophic forgetting?

(2). The reconstruction module currently uses only simple convolutional and activation layers. If more complex layers were introduced in this module, would the performance improve?

(3). Does the performance improve if the contrastive loss term L_CR is used independently, without the reconstruction module?

(4). There is a typo in Table 1. The best result of Inc 5 experiment on CIFAR100 B50 dataset is obtained by DSGD (63.58) instead of CREATE (63.53).

(5). During ablation experiments, how are the prototypes in NCM obtained? Do they update with tasks? If only prototypes obtained from previous tasks are used without any updating strategy, severe forgetting is inevitable.

---

### Note · Authors · 2024-11-15

I have read and agree with the venue's withdrawal policy on behalf of myself and my co-authors.